# Findings of Electrodiagnostic Studies in Moderate to Severe Lumbar Central Spinal Stenosis—Electrodiagnostic Studies in Lumbar Central Spinal Stenosis

**DOI:** 10.3390/healthcare9020164

**Published:** 2021-02-03

**Authors:** Min Cheol Chang, Donghwi Park

**Affiliations:** 1Department of Rehabilitation Medicine, College of Medicine, Yeungnam University, Daegu 42415, Korea; wheel633@ynu.ac.kr; 2Department of Physical Medicine and Rehabilitation, Ulsan University Hospital, University of Ulsan College of Medicine, 877, Bangeojinsunghwando-ro, Dong-gu, Ulsan 44033, Korea

**Keywords:** lumbar spine, spinal stenosis, electrodiagnostic study, nerve conduction study, electromyography

## Abstract

*Purpose*: The purpose of this study was to investigate the findings of electrodiagnostic studies (nerve conduction study (NCS) and electromyography (EMG)) in patients with moderate and severe lumbar central spinal stenosis (LCSS). *Methods*: We retrospectively reviewed the medical records of Ulsan University Hospital and identified 32 consecutive patients (mean age = 66.9 ± 7.4 years; male:female = 8:24) with LCSS. Based on the results of T2 axial magnetic resonance imaging at the level of L4–5, patients were categorized as having severe (*n* = 14) or moderate LCSS (*n* = 18). Results from NCS and EMG were retrieved. Additionally, we included 15 age- and sex-matched volunteers without LCSS (mean age = 65.2 ± 8.0 years; male:female = 4:11) to serve as a control group. Results of NCS and EMG were compared between the three groups. *Results*: We found that, compared to normal subjects, patients with moderate or severe LCSS presented significantly lower distal amplitudes of the compound motor action potential of both peroneal and tibial nerves. Regarding EMG, positive sharp waves and fibrillation potentials were exclusively observed in patients with severe LCSS group (28.6%). *Conclusion*: Electrodiagnostic studies were significantly altered in patients with moderate and severe LCSS. Our results may be helpful to diagnose LCSS-induced radiculopathy and to differentiate it from other causes of peripheral nerve pathologies.

## 1. Introduction

Lumbar central spinal stenosis (LCSS) is defined as the narrowing of the lumbar spinal canal due to bulging intervertebral discs and/or hypertrophy of the ligamentum flavum and facet joints that results in the compression of nerve roots [1]. Being a degenerative process associated with age, it predominantly affects individuals older than 50 years [2]. Symptomatic LCSS affects approximately 27% of the general population and represents one of the leading causes of visits to pain clinics [3,4].

The most characteristic symptom of LCSS is neurogenic claudication, which refers to leg pain, fatigue, heaviness, and/or weakness that typically worsens with lumbar extension [5]. The diagnosis of LCSS relies on a combination of symptoms, physical findings, and imaging study results (most commonly magnetic resonance imaging (MRI) and computed tomography (CT) [5,6]. Additionally, electrodiagnostic studies such as nerve conduction study (NCS) and electromyography (EMG) are often used to identify the specific site to be treated when equivocal findings and/or multiple-level lesions are detected via CT or MRI [5]. The typical electrophysiological finding in LCSS is radiculopathy, which results from nerve root damage by mechanical compression or ischemic injury [7,8]. Although NCS and EMG are commonly used for detecting radiculopathy in patients with spinal stenosis, their sensitivity is thought to be low [9,10,11,12,13]. However, studies addressing this issue have included patients with a wide spectrum of stenosis, ranging from mild to most severe. In this regard, we consider that moderate to severe forms of LCSS may have a higher incidence of radiculopathy, and in this setting, the role of electrodiagnostic studies may be particularly important.

Therefore, we aimed to investigate the findings of NCS and EMG in patients with moderate and severe LCSS and compare them with electrodiagnostic results of patients without LCSS. Additionally, we aimed to compare findings between patients with moderate and severe LCSS.

## 2. Methods

### 2.1. Study Design and Patient Selection

This study was approved by the Institutional Review Board/Ethics Committee of Yeungnam University Hospital, informed consent was waived because of the retrospective nature of the study, and the analysis used anonymous clinical data. In this retrospective study, we included 32 consecutive patients (mean age = 66.9 ± 7.4; male:female = 8:24) who visited the spine center of Yeungnam University Hospital from January 2014 to December 2019. Patients were considered for analysis if they met all of the following criteria: (1) pain attributable to LCSS, characterized by buttock and/or lower extremity pain that appeared during walking or prolonged standing and was relieved by leaning forward or sitting; (2) moderate or severe LCSS diagnosed on axial MRI, as explained below; (3) age between 55 and 79 years; (4) NCS and EMG conducted at the lower extremity; and (5) most severe degree of stenosis at level L4–5. Exclusion criteria were as follows: (1) severe foraminal stenosis, lumbar disc herniation, myelopathy, or spine infection; (2) history of spinal surgery, such as lumbar fusion or laminectomy; (3) history of cancer; (4) diabetes; (5) history of peripheral neuropathy; and (6) symptoms of distal symmetric polyneuropathy (distal neuropathic pain at rest).

To serve as control, NCS and EMG were performed in 15 age- and sex-matched normal volunteers (mean age = 65.2 ± 8.0; M:F = 4:11) who had no symptoms of lumbar stenosis, lumbar disc herniation, or distal symmetric polyneuropathy and no history of spinal surgery, cancer, diabetes, or peripheral neuropathy (Table 1). The study protocol was approved by the Institutional Review Board of Yeungnam University Hospital. Informed consent was obtained from all volunteers in the normal group and was waived for LCSS patients due to the retrospective nature of the study.

The severity of LCSS was assessed on MRI at the L4–5 level, based on the grading system proposed by Lee et al. (Figure 1) [14]. Grade 0 corresponded to no LCSS; grade 1 to mild stenosis, with clear separation of each cauda equina nerve root; grade 2 to moderate stenosis, with some cauda equina aggregation; and grade 3 to severe stenosis, with the entire cauda equina appearing as a single bundle.

### 2.2. Electrodiagnostic Studies

NCS and EMG were conducted by a single technician examiner using a Nicolet EDX system and Viking software (CareFusion 209 Inc., Middleton, WI, USA). We collected results of motor NCS of peroneal and tibial nerves and EMG data on paraspinal and lower extremity muscles. Studies were performed on the lower extremity that exhibited the most severe pain. When symptom severity was comparable between extremities, the examiner randomly decided which side to test.

In peroneal motor NCS, the recording electrode was placed on the extensor digitorum brevis, and the reference electrode was placed at the base of the fifth toe, respectively. Stimulation of the peroneal nerve was applied on the ankle, lateral to the anterior tibialis tendon, and just below the fibula head. The ground was placed on the dorsum of the foot. In tibial NCS, the recording and reference electrodes were placed over the abductor hallucis (1 cm below and behind the navicular tubercle) and the base of the first toe, respectively. The ground was placed over the dorsum of the ankle. Stimulation of the tibial nerve was applied distally posteriorly to the medial malleolus and proximally at the level of the knee in the lower border of the popliteal space near the popliteal artery.

In motor NCSs, the distal amplitude, distal latency, and conduction velocity of the compound motor action potential (CMAP) were collected. For the peroneal nerve, cut-off values for normal distal amplitude, distal latency, and conduction velocity were set at 1.3 mV, 6.5 ms, and 38 m/s, respectively; for the tibial nerve, the respective cut-off values were 4.4 mV, 6.1 ms, and 39 m/s [15]. EMG was evaluated on the following muscles: iliopsoas, vastus medialis, tibialis anterior, peroneus longus, tensor fascia latae, and medial head of gastrocnemius. If necessary, the long head of biceps femoris and gluteus maximus were also assessed. A 50-mm disposable, concentric needle was used. Abnormal spontaneous activities were assessed with a gain of 50 μV, sweep speed of 10 ms/division, and filter settings of 10 Hz to 10 kHz. Voluntary activity was assessed with a gain of 200–500 mV, sweep speed of 10 ms/division, and filter settings of 10 Hz to 10 kHz.

### 2.3. Statistical Analysis

All statistical analyses were performed using the Statistical Package for the Social Sciences software (SPSS for Windows version 23.0, IBM corp., Armonk, NY, USA). Due to the small and variable sizes of the three study groups (severe, moderate, and normal groups), we used the non-parametric one-way ANOVA test (the Kruskal–Wallis test with the Mann–Whitney *U* test) for comparison of demographic and NCS data. Additionally, we used the chi-square test to compare nominal (categorical) data between groups. In cases where more than 20% of cells had a frequency <5, we used the Fisher’s exact test. Statistical significance was set at *p* < 0.05.

## 3. Results

Demographic and electrodiagnostic results are shown in Table 1. The demographic characteristics (age and sex ratio) were not significantly different between the three groups (severe, moderate, and normal groups).

Regarding NCS, while distal amplitudes in both the peroneal and tibial nerves were significantly smaller in the severe and moderate LCSS groups compared to the normal group, no significant difference was observed within the LCSS groups in this regard. Additionally, distal latency and conduction velocity values were comparable between the three groups, regardless of the specific nerve assessed (Table 1). Similarly, no significant difference was found between groups regarding the incidence of abnormal distal amplitudes (< 1.1 mV) of the peroneal nerve (Fisher’s exact test, *p* = 0.298). One patient in the severe group showed a peroneal distal amplitude lower than the cut-off value.

Regarding EMG results, abnormal spontaneous activities were evidenced only in the severe group (Table 1). More specifically, four patients (28.6%) showed positive sharp waves and fibrillation potentials (scores of 1+ or 2+ in all cases) on the muscles innervated by L5.

## 4. Discussion

This study was the first to compare electrodiagnostic findings between patients with moderate or severe LCSS and normal subjects. In addition, we compared the results of an electrodiagnostic study of patients with severe LCSS with those of patients with moderate LCSS. As our results show, patients with moderate or severe LCSS showed significantly lower distal amplitudes in peroneal and tibial NCSs, compared to subjects without LCSS. Similarly, abnormal spontaneous activities (positive sharp waves and fibrillation potentials) were exclusive to severe LCSS and affected a significantly large proportion of patients in this group (4/14, 28.6%).

In lumbar stenosis, the narrowing of the spinal canal can result in direct or indirect mechanical compression of nerve roots [16]. Additionally, the rise in intrathecal pressure can compromise venous and arterial blood flow, leading to ischemic injury of lumbosacral nerve roots and further compromising impulse conduction [17]. As the nerve root injury due to LCSS is a preganglionic lesion, the dorsal root ganglion is intact. Therefore, sensory nerve conduction is spared, and abnormal findings can be found on only motor nerve conduction.

L5 is the most frequently affected nerve root in LCSS. Peroneal and tibial nerves receive a significant contribution of fibers from this root and thus constitute a suitable target for NCS [18]. Since LCSS provokes a focal injury on the nerve, Wallerian degeneration can occur distally, which seems to contribute to the lowered amplitude of CMAPs of peroneal and tibial nerves [19]. Although LCSS was associated with lower distal amplitudes in this study, it is important to note that the average values of CMAPs of peroneal and tibial nerves were not decreased to a pathological (severe) degree. This may be due to the gradual nature of nerve root injury in LCSS, which may be due to the possibility to compensate for some serious damage due to the repeated course of nerve root injury and recovery. [20,21] Furthermore, neovascularization and development of a collateral blood supply in the spinal canal could also contribute to mitigating injury [22].

In patients with radiculopathy, positive sharp waves and fibrillation potentials appear in paraspinal and the corresponding myotome limb muscles due to membrane instability following axonal loss [23]. In our study, these findings were exclusive of patients with severe LCSS. In this regard, we speculate that axonal damage in patients with moderate LCSS might not be significant enough to cause electromyographic alterations.

Results from other studies are generally in accordance with ours. For example, Haig et al. [10] conducted electrodiagnostic studies in 24 patients with lumbar stenosis and reported that lower extremity fibrillation potentials were observed in 33.3% of cases. Similarly, Egli et al. [24] performed motor NCSs in 54 patients with LCSS who were scheduled for surgery and found abnormal CMAPs of the posterior tibial nerve (defined as a distal amplitude < 5.7 mV) in 39% of their cohort. Despite differences in baseline patient characteristics and cut-off values, these results and ours agree in showing that LCSS is associated with abnormal CMAP results. Regarding the relation between the grade of stenosis and electrodiagnostic findings, a recent study including 115 patients found no relation between the severity of LCSS and the results of NCS on the lower extremity [10]. Although this is consistent with our results, the control group did not include normal subjects. We consider that this study failed to evaluate NCS findings in LCSS.

Our study has some limitations. First, patients with LCSS were recruited retrospectively. Second, the number of subjects included was small. Third, F- and H-waves were not evaluated. Last, we did not include the height of the subjects as an additional variable. Further studies compensating our limitations should be conducted in the future.

In conclusion, we found that moderate and severe LCSS were associated with a lowered CMAP distal amplitude in NCS. Additionally, positive sharp waves and fibrillation potentials were exclusively present in severe stenosis, affecting almost 30% of patients. We think that an electrodiagnostic study can be useful for evaluating the degree of nerve root damage by spinal stenosis. Furthermore, we consider that our results may be helpful to diagnose radiculopathy due to LCSS and to differentiate it from other peripheral nerve pathologies.

## Figures and Tables

**Figure 1 healthcare-09-00164-f001:**
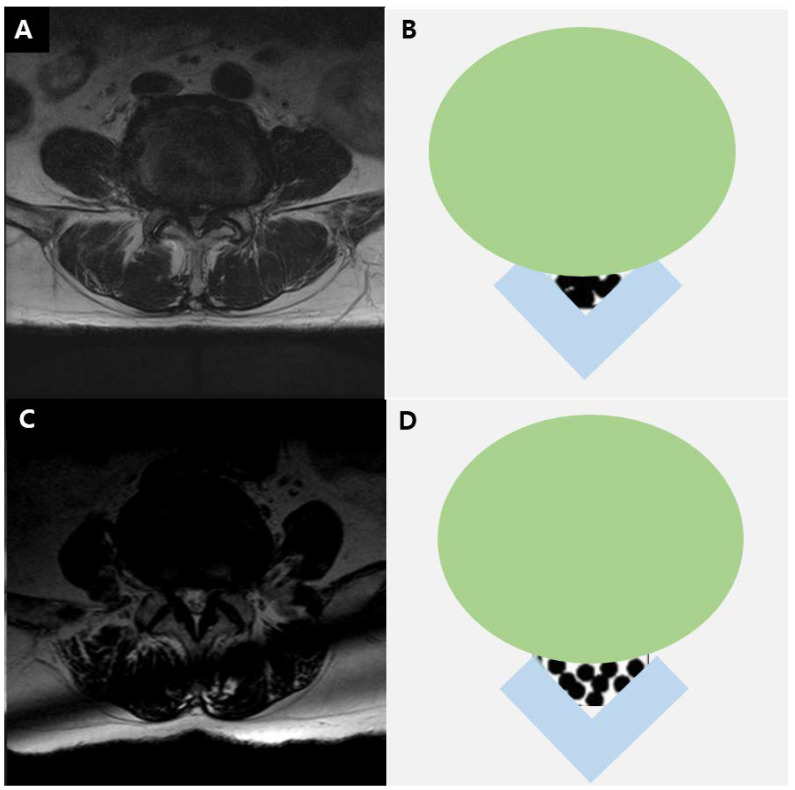
T2-weighted axial magnetic resonance image in a 73-year-old man (**A**) and diagram (**B**) showing severe lumbar stenosis. T2-weighted axial magnetic resonance image in a 70-year-old man (**C**) and diagram (**D**) showing moderate lumbar stenosis.

**Table 1 healthcare-09-00164-t001:** Demographic and electrodiagnostic data.

	Severe Group	Moderate Group	Normal Group	*p*-Value
*n* (male sex)	14 (4)	18 (4)	15 (4)	0.919
Mean age ± SD, years	66.9 ± 8.6	66.9 ± 6.7	65.2 ± 8.0	0.865
CMAP, peroneal nerve				
Mean distal amplitude ± SD, mV	4.8 ± 2.2	6.0 ± 2.3	8.2 ± 2.0	α: 0.283, β: < 0.001 *, γ: 0.004 *
Mean distal latency ± SD, ms	4.0 ± 0.5	4.3 ± 0.6	4.0 ± 0.5	0.340
Mean velocity ± SD, m/s	46.9 ± 4.8	45.1 ± 3.1	44.0 ± 3.3	0.141
CMAP, tibial nerve				
Mean distal amplitude ± SD, mV	19.7 ± 7.3	18.3 ± 5.0	25.6 ± 7.8	α: 0.722, β: < 0.001 *, γ: 0.041 *
Mean distal latency ± SD, ms	3.9 ± 4.4	4.1 ± 0.7	3.9 ± 4.4	0.594
Mean velocity ± SD, m/s	46.9 ± 3.2	44.7 ± 3.6	45.9 ± 2.5	0.123
EMG				
Positive sharp waves and fibrillation potentials, *n*	4	0	0	α: 0.028 *, β: 0.042 *, γ: 1.000

SD: standard deviation. CMAP: compound motor action potential. EMG: electromyography; α: severe group vs. moderate group; β: severe group vs. normal group; γ: moderate group vs. normal group; * *p*-value < 0.05.

## Data Availability

The data presented in this study are available on request from the corresponding author. The data are not publicly available due to privacy restriction.

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
