# Peer review of "Findings of Electrodiagnostic Studies in Moderate to Severe Lumbar Central Spinal Stenosis—Electrodiagnostic Studies in Lumbar Central Spinal Stenosis"

_healthcare, 2021, doi:10.3390/healthcare9020164_

Round 1
Reviewer 1 Report
This article presents the significance of an assessment of several moderate to severe degrees of spinal stenosis by MRI imaging, supported by electrodiagnostic tests such as nerve conduction testing (NCS) and electromyography (EMG).
To authors.
Please put the data in the appropriate lines in table 1
Please complete the EMG methodology in more detail.
The correlation between spinal stenosis and electrodiagnostic results can be used to support the importance of the additional electrodiagnostic tests used.
Author Response
Reviewer 1
This article presents the significance of an assessment of several moderate to severe degrees of spinal stenosis by MRI imaging, supported by electrodiagnostic tests such as nerve conduction testing (NCS) and electromyography (EMG).
To authors.
Please put the data in the appropriate lines in table 1
Answer: I appreciate your comments. We put the data in the appropriate line.
Please complete the EMG methodology in more detail.
Answer: I appreciate your comments. We described the methodology of the electrodiagnostic study including EMG in more detail.
The correlation between spinal stenosis and electrodiagnostic results can be used to support the importance of the additional electrodiagnostic tests used.
Answer: We added the contents on the usefulness of EMG on evaluating the degree of severity of nerve root damage to the discussion section.
Reviewer 2 Report
Please see the comments in the attached manuscript.

Author Response
we submitted a response for reviewer`s comment as a PDF form.

Round 2
Reviewer 1 Report
The authors have corrected the indicated parts of the article.
This is the basis for accepting the article to the publication in the present form.
The authors have completed the methodology and emphasized the importance of additional diagnostics in cases of moderate and acute spinal stenosis in the discussion section.